# Uptake and Effectiveness of Risk-Reducing Surgeries in Unaffected Female *BRCA1* and *BRCA2* Carriers: A Single Institution Experience in the Czech Republic

**DOI:** 10.3390/cancers15041072

**Published:** 2023-02-08

**Authors:** Martina Zimovjanova, Zuzana Bielcikova, Michaela Miskovicova, Michal Vocka, Anna Zimovjanova, Marian Rybar, Jan Novotny, Lubos Petruzelka

**Affiliations:** 1Department of Oncology, First Faculty of Medicine, Charles University and General University Hospital, U Nemocnice 499/2, 128 08 Prague, Czech Republic; 2Department of Oncology, Nitra Faculty Hospital, Špitálska 6, 949 01 Nitra, Slovakia; 3Faculty of Medicine, Masaryk University, 601 77, Kamenice 5, 625 00 Brno, Czech Republic; 4International Clinical Research Center (ICRC) of St Anne’s University Hospital (FNUSA), Pekařská 664/53, 602 00 Brno, Czech Republic; 5Department of Biomedical Technology, Faculty of Biomedical Engineering, Czech Technical University, náměstí Sítná 3105, 272 01 Kladno, Czech Republic; 6Department of Surgery, Sunderby Hospital, Sjukhusvägen 10, 954 42 Sunderbyn, Sweden

**Keywords:** *BRCA1*, *BRCA2*, breast cancer, ovarian cancer, risk-reducing mastectomy, risk-reducing salpingo-oophorectomy, cancer prevention, Angelina Jolie effect

## Abstract

**Simple Summary:**

Women with *BRCA 1/2* pathogenic/likely pathogenic variants (P/LPVs) have a high lifetime risk of developing breast and ovarian cancer. The aim of the retrospective study is to analyze the rate, longitudinal trends, and effectiveness of prophylactic risk-reducing mastectomy (RRM) and salpingo-oophorectomy (RRSO) on the incidence of breast and ovarian cancer. We analyzed data from 496 unaffected *BRCA1/2* carriers with a median follow-up of 6.0 years. A statistically significant increase of RRM (12% vs. 29%) and RRSO (31% vs. 42%) was observed when comparing periods 2005–2012 and 2013–2020 (*p* < 0.001). BC developed in 15.9% of *BRCA1/2* carriers without RRM vs. 0.6% of *BRCA1/2* carriers after RRM (HR 20.18, *p* < 0.001). OC was diagnosed in 4.3% vs. 0% of *BRCA1/2* carriers without vs. after RRSO (*p* < 0.001). Data shows a high effectiveness and significant increase of prophylactic surgeries over 20 years period in a central European population of *BRCA1/2* carriers.

**Abstract:**

Unnafected female carriers of *BRCA1* and *BRCA2* pathogenic/likely pathogenic variants (P/LPVs) are at higher risk of breast cancer (BC) and ovarian cancer (OC). In the retrospective single-institution study in the Czech Republic, we analyzed the rate, longitudinal trends, and effectiveness of prophylactic risk-reducing mastectomy (RRM) and risk-reducing salpingo-oophorectomy (RRSO) on the incidence of BC and OC in *BRCA1*/*2* carriers diagnosed between years (y) 2000 to 2020. The study included 496 healthy female *BRCA1*/*2* carriers. The median follow-up was 6.0 years. RRM was performed in 156 (31.5%, mean age 39.3 y, range 22–61 y) and RRSO in 234 (47.2%, mean age 43.2 y, range 28–64 y) *BRCA1*/*2* carriers. A statistically significant increase of RRM (from 12% to 29%) and RRSO (from 31% to 42%) was observed when comparing periods 2005–2012 and 2013–2020 (*p* < 0.001). BC developed in 15.9% of *BRCA1/2* carriers without RRM vs. 0.6% of *BRCA1/2* carriers after RRM (HR 20.18, 95% CI 2.78- 146.02; *p* < 0.001). OC was diagnosed in 4.3% vs. 0% of *BRCA1/2* carriers without vs. after RRSO (HR not defined due to 0% occurrence in the RRSO group, *p* < 0.001). Study results demonstrate a significant increase in the rate of prophylactic surgeries in *BRCA1/2* healthy carriers after 2013 and the effectiveness of RRM and RRSO on the incidence of BC and OC in these populations.

## 1. Introduction

Hereditary pathogenic/likely pathogenic variants (P/LPVs) of *BRCA1* and *BRCA2* genes are the leading genetic causes of breast cancer (BC), ovarian cancer (OC), and other cancers (prostate, pancreas). The hereditary background of *BRCA1/2* pathogenic alterations is estimated in 5–10% of all BC cases and 8–13% of all OC cases [1,2,3]. Cumulative BC risk at 80 years is 72% for *BRCA1* and 69% for *BRCA2* carriers [4,5,6]. The lifetime OC risk is 44% in *BRCA1* and 17% in *BRCA2* carriers [4,5,6]. The proportion of *BRCA1* vs. *BRCA2* is specific to different populations in various regions, with the predominant *BRCA1* incidence in the Czech Republic [7].

All *BRCA1/2* carriers without previous history of breast or ovarian cancer are encouraged to participate in a specific preventive program and recommended to consider risk-reducing surgeries at the oncological institutions in the Czech Republic. The Czech Society of Clinical Oncology regularly updates guidelines on a surveillance program and preventive strategy for *BRCA1/2* carriers. The surveillance program consists of a clinical and radiological examination beginning at age 21. Breast magnetic resonance imaging is done every twelve months for carriers at age 25–75. Mammogram (MMG) is performed every twelve months for carriers at age 30–75. The radiological screening methods are combined in six months intervals.

During the surveillance program, oncologists frequently consult prophylactic procedures with *BRCA 1/2* carriers (risk-reducing mastectomy RRM and risk-reducing salpingo-oophorectomy RRSO). RRM is the elective surgical removal of both breasts to prevent future breast cancer. RRSO comprises bilateral adnexectomy with SEE-FIM protocol and lavage with cytology of the abdominal cavity. The SEE-FIM protocol (pathology dissection protocol for Sectioning and Extensively Examining the FIMbria) has been used in the Czech Republic since 2008. The protocol was designed to provide the optimal microscopic examination of the distal fallopian tube (fimbria) to identify either cancerous or precancerous conditions in this organ [8].

Breast screening using an annual MRI combined with an MMG is important for detecting BC at an early and likely curable stage [9,10,11]. OC screening with transvaginal ultrasound (TVUS) and blood test for cancer antigen 125 (CA125) are not sufficiently effective in the early detection of the disease [12].

A systematic review of twenty-one observational studies in healthy *BRCA1/2* carriers describes that risk-reducing mastectomy (RRM) effectively reduces both the incidence and mortality from BC [13]. However, other systemic reviews also emphasize the need for more rigorous prospective studies to confirm the survival benefits of RRM in *BRCA1/2* carriers [14,15]. The impact of risk-reducing salpingo-oophorectomy (RRSO) on BC risk reduction is still uncertain in the present literature [16]. It varies depending on the age, type of *BRCA1, BRCA2*, and history of exposure to female hormones [17,18]. RRSO decreases the incidence and mortality of OC and the incidence of the fallopian tube and primary peritoneal cancer. RRSO is recommended once the desire for pregnancy is completed in women with BRCA1 P/LPVs from the ages of 35 to 40 years, and in women with BRCA2 P/LPVs from the ages of 40 to 45 years onwards [9,10].

The proportion of healthy *BRCA1/2* carriers undergoing prophylactic surgeries (RRM, RRSO) varies worldwide [19,20,21]. A long-term analysis of the uptake of RRM describes significant differences in the uptake of risk-reduction strategies in ten countries [22]. Another finding from a published analysis of Metcalfe et al. was a significant increase in breast MRI screening and an increase in uptake of RRM in the period 2009–2017 compared to 1995–2008. 

Our study aims to assess the rate of prophylactic surgeries, timing, and longitudinal trends of uptake of RRM and RRSO in healthy *BRCA1/2* carriers in the Czech Republic. We analyze the impact of risk-reducing surgeries on BC and OC incidence in twenty years, 2000–2020. The study also describes additional data regarding the histopathology and type of treatment of detected BC and OC.

## 2. Materials and Methods

### 2.1. Study Setting

Genetic testing for *BRCA1/2* P/LPVs was conducted from 1998 to 2020 at the Laboratory of Oncogenetics, First Faculty of Medicine, Charles University, Prague, Czech Republic. A specific surveillance program for healthy *BRCA1/2* carriers started in 2000 at the Department of Oncology, General University Hospital Prague, Czech Republic. According to updated international guidelines, the specialists offer *BRCA1/2* carriers screening strategies for early BC and OC detection and counsel the benefits of risk-reducing surgeries (RRM, RRSO) [9,10]. 

The primary endpoint of this retrospective study was to evaluate the number and timing of RRM and RRSO. Time to prophylactic surgery (TTPS) was defined as the time from initiating the surveillance program to undergoing RRM/RRSO. We calculated the median and the mean age at the time of prophylactic surgery. The secondary endpoint was to assess the incidence of BC and OC in *BRCA1/2* carriers during the study period from 2000 to 2020. *BRCA1/2* carriers with a history of BC, OC, or other cancers at the time of genetic testing for P/LPVs were excluded from the study. *BRCA1/2* carriers with BC developed during the surveillance program were included in the analysis regarding the effectiveness of RRSO on OC/FTC incidence. *BRCA 1/2* carriers with BC developed during the surveillance program were included in the "non-RRM group" in the analysis regarding the effectiveness of RRM on BC incidence. *BRCA 1/2* carriers with secondary breast cancer or contralateral prophylactic mastectomy were excluded from the dataset. We calculated the effectiveness of RRM on the incidence of BC and the effectiveness of RRSO on the incidence of OC separately.

Subjects with incomplete medical records or loss of follow-up were excluded. The following information was collected: type of *BRCA1* and *BRCA2* pathogenic variants, the age of women at the time of genetic diagnosis, the age at the time of prophylactic surgery, and the age at diagnosis of BC/OC cancer. The percentage of healthy *BRCA1/2* carriers who underwent RRM, RRSO, or combined surgical procedures was counted. 

Follow-up was defined as the time from initiating the surveillance program until the end of the data collection. BC/OC was detected in three different clinical scenarios: before prophylactic surgery (PS), at the time of PS, and after PS. Diagnosis of occult cancer was based on the histopathological report of the patient’s tissue after RRM/RRSO. Patients with occult breast cancer (DCIS or invasive BC diagnosed based on histopathological findings after RRM) were included in the cohort of *BRCA1/2* carriers without previous RRM. Women with occult ovarian cancer (STIC or invasive OC/FTC diagnosed based on histopathological findings after RRSO) were included in the cohort of *BRCA1/2* carriers without previous RRSO.

### 2.2. Molecular Analysis

An analysis of P/LPVs in *BRCA1/2* genes was initially (2000–2014) performed by protein truncation test or direct sequencing. For the analysis of the presence of large genomic *BRCA1/2* rearrangements, multiplex ligation-dependent probe amplification was used (MRC Holland, Amsterdam, the Netherlands) [23,24,25]. Since 2015, all samples have been analyzed using the custom-designed CZECANCA panel (NimbleGen/Roche, Pleasanton, CA, USA) targeting 219 cancer-susceptible genes on MiSeq (Illumina, San Diego, CA, USA) [26]. The bioinformatics analysis included the identification of pathogenic variants (single nucleotide variants described as pathogenic in ClinVar, non-sense, frame-shift, splicing-site alterations, and copy number variants) using a pipeline described in the previous text [26,27].

### 2.3. Statistical Analysis

Statistical analysis was performed using the software Statistica 13 (Tulsa, OK, USA) and software R. Continuous variables were reported as means and range. Two-sample *t*-tests were used to compare age in groups of *BRCA1* and *BRCA2* carriers. Categorical variables were reported as proportions. Pearson’s chi-squared tests were used to compare groups without and with surgical intervention (RRM, RRSO) and periods 2005–2013 vs. 2013–2020. The Kaplan-Meier curves with Hazard ratios (HR) and Log-rank tests were used to compare the time to BC and OC occurrence between groups of *BRCA1/2* carriers without and with surgical intervention. We used the right censoring in the study. Censored were patients who had reached the end of follow-up but had not yet experienced an event (breast and ovarian cancer). All tests were performed at the 5% level of significance.

## 3. Results

In this single-institution retrospective analysis (2000–2020), we enrolled 1191 carriers of *BRCA1/2* P/LPVs. Individuals assigned as males at birth were excluded from the study since RRM is recommended only for female carriers. Out of 940 *BRCA1/2* females (assigned at birth), 431 had a prior history of some cancer at the time of genetic diagnosis, and 13 *BRCA1/2* carriers had incomplete medical records or had lost follow-up. These women were excluded from the study. Finally, 496 unaffected/healthy female *BRCA1/2* carriers met the eligibility criteria and were included in the analysis, as is seen in Figure 1. 

### 3.1. BRCA1/2 Carriers Surveillance and Prophylactic Procedures 

From the total count of 496 carriers, 348 women (70.2%) were positive for *BRCA1,* and the remaining 148 women (29.8%) were positive for *BRCA2*. The median follow-up of the study was 6.0 years. The mean age of *BRCA1/2* carriers at the initiation of the follow-up was 35.4 years (18–65 y). 

Out of 496 healthy *BRCA1/2* carriers, 156 (31.5%) underwent RRM (53 women RRM only, 103 women combination of RRM and RRSO). The mean age for the whole group of *BRCA1/2* carriers at the time of RRM was 39.3 years. A subgroup of 53 healthy carriers who decided on RRM only procedure had a mean age of 32.9 years, without significant differences between groups of *BRCA1* and *BRCA2* carriers. From 156 women exposed to RRM, 126 (80.8%) healthy *BRCA1/2* carriers had breast reconstructive surgery. RRSO underwent 234 (47.2%) *BRCA1/2* carriers (131 women RRSO only, 103 women combination of RRM and RRSO). The mean age of *BRCA1/2* carriers at the time of RRSO was 43.2 years, also without significant differences between groups of *BRCA1* and *BRCA2* carriers. 

Both groups of *BRCA1* and *BRCA2* carriers were almost equally willing to undergo prophylactic surgeries. RRM was undertaken in 111 out of 348 (31.9%) *BRCA1*-positive women and 45 out of 148 (30.4%) *BRCA2*-positive women. Similarly, 162 out of 348 (46.6%) *BRCA1* vs. 72 out of 148 (48.6%) *BRCA2* carriers underwent RRSO. A total of 103 (20.8%) *BRCA1/2* women underwent both types of procedures; RRSO was preferred by patients as the first surgery. 65 *BRCA 1/2* carriers preferred RRSO as the first prophylactic procedure and 38 *BRCA 1/2* carriers preferred RRM first.

The remaining 209 (42.1%) *BRCA1/2* carriers were only under surveillance program. The surveillance program for women without RRSO consisted of screening examinations every six months (transvaginal ultrasound TVUS and blood test CA125). A summary of the data is shown in Table 1. 

### 3.2. Changes in Uptake of Risk-Reducing Surgeries over the Time 

An upward trend in the number of prophylactic surgeries (RRM, RRSO) was observed during the study. From 2000 to 2005, no prophylactic surgeries were performed on healthy *BRCA1/2* carriers. Contrary, from 2005 to 2020, prophylactic surgeries were performed in increasing numbers. A statistically significant increase in the rate of RRM uptake was found between the periods 2005–2012 vs. 2013–2020 (*p* < 0.001). 2013 was a crucial time when the uptake of prophylactic surgeries started to rise significantly higher rate than in previous years worldwide.

From 2005 to 2012, 23 out of 192 (12%) newly recruited healthy *BRCA1/2* carriers underwent RRM. From 2013 to 2020, the number of *BRCA1/2* carriers undergoing RRM has more than doubled and 96 out of 304 (31.6%) newly recruited healthy *BRCA1/2* carriers were exposed to RRM. 

Similarly, a significant difference in RRSO uptake has been described since 2013. The number of operated women increased from 56/192 (29.2%) to 129/304 (42.4%) in comparison of periods 2005–2012 vs. 2013–2020 (*p* < 0.001).

Time to prophylactic surgery (TTPS) measured from initiating of the surveillance program was longer in the period 2005–2012 (time to RRM 2.3 y and time to RRSO 1.5 y) than TTPS in the period 2013–2020 (time to RRM 1.8 y and time to RRSO 1.0 y). However mean age at the time of prophylactic surgery was lower in 2005–2012 than in 2013–2020 (39.2 y vs. 40.2 y for RRM, *p* = 0.58, non-significant; and 41.3 y vs. 44.9 y for RRSO, *p* = 0.004, significant). Details are also seen in Table 2.

### 3.3. Incidence of BC in BRCA1/2 Carriers with/without RRM and BC Treatment Strategy 

Of 496 healthy *BRCA1/2* carriers, 55 (11.1%) BC were detected during the study period. In the cohort of 348 *BRCA1* carriers, 46 (13.2%) BC were detected. Out of 148 *BRCA2* carriers, nine women (6.1%) developed BC. The mean age of BC diagnosis was 39.6 y (39.5 y for *BRCA1* and 42.3 y for *BRCA2*). 

BC was significantly more frequent in the subset of women without RRM (HR 20.18, *p* < 0.001), as seen in Figure 2. 54 BC (15.9%) was diagnosed in a group of 340 women without previous RRM and only one (0.6%) out of 156 women after prophylactic RRM. 

Basic clinical and pathological characteristics of BC diagnosed in *BRCA1/2* carriers are described in Table 3. The main histological subtype was invasive ductal carcinoma in both groups of patients (89.1% total). The predominant subtype was triple-negative BC (TNBC) in *BRCA1* carriers (63.1%) and estrogen receptor-positive (ER-positive) BC in *BRCA2* carriers (88.9%). TNBC was more commonly associated with high grade (71.8%) compared to ER-positive BC (44.4%). Similarly, the distribution of BC stages was more favorable for ER-positive than for TNBC (55.6% vs. 47.8% of patients in stage I). Three of five cases of DCIS were diagnosed at the time of RRM and were described as occult cancer in final pathological reports. Only 58.2% of women were diagnosed with early-stage BC. Diagnosis in an early stage of BC is the goal of the surveillance program. Vice versa, no stage III or IV BC was detected. 42 BC cases were diagnosed during the regular radiological screening (screen-detected BC). MMG and MRI are combined in six months intervals. 13 BC cases were diagnosed during inter-screening intervals after the previous negative radiological examination (interval BC).

The treatment strategy of BC patients was selected based on the stage and disease characteristics and is described in Table 3. The sequence of neoadjuvant chemotherapy (CT) followed by surgery was more common in *BRCA1* carriers; in *BRCA2* carriers was more often chosen the opposite sequence. Five patients with DCIS were fully treated with surgery. Only five women (9.1%) with invasive BC were treated with endocrine therapy (ET) without CT. The extent of BC surgery was determined based on disease stage and patient preferences. Of 46 *BRCA1*-positive patients, 29 (63%) underwent a bilateral mastectomy. Vice versa, six out of nine (66.7%) *BRCA2*-positive patients preferred breast-conserving surgery.

### 3.4. Incidence of OC in BRCA1/2 Carriers with/without RRSO and OC Treatment Strategy 

Ovarian cancer is a group of diseases that originates in the ovaries or the related areas of the fallopian tubes and the peritoneum. Out of 496 healthy BRCA1/2 carriers, eleven (2.2%) OC were detected (four serous tubal intraepithelial cancers and seven invasive ovarian/fallopian tube cancers). Nine (2.5%) OC occurred in 348 BRCA1 carriers and two (1.4%) among 148 BRCA2 carriers. The mean age of patients at the time of OC diagnosis was 52.1 years (51.2 y for BRCA1 and 53.5 y for BRCA2). The statistically significant difference in the incidence of OC in BRCA1/2 carriers, without and with the intervention of RRSO, was counted (HR not defined due to 0% occurrence of OC in the RRSO group, p < 0.001), as seen in Figure 3. In the group of 262 healthy BRCA1/2 carriers without previous RRSO, eleven (4.2%) OC was detected. Contrary, no case of OC was diagnosed after RRSO. RRSO procedures have been conducted according to the SEE-FIM protocol since 2008 in the Czech Republic. 29 of 234 RRSOs were conducted before 2008 without SEE-FIM protocol. All cases of STICs and invasive cancers were detected between 2012–2019 (with SEE-FIM protocol).

The basic characteristics, diagnosis, and treatment of OC in *BRCA1/2* carriers are described in Table 4. Serous tubal intraepithelial cancer (STIC) is a lesion limited to the fallopian tube epithelium and a precursor to extrauterine (pelvic) high-grade serous cancer [28]. Four STICs were detected from RRSO pathological findings. CA125 and transvaginal ultrasound (TVUS) were normal in these cases before the surgery. The surgical procedures had curative effects on the *BRCA1* carriers. With the medium follow-up of 4.7 years, no recurrence nor primary peritoneal serous tumor was detected.

Invasive ovarian/fallopian tube cancer is classified according to TNM or FIGO classifications. Seven cases of invasive cancer were detected in *BRCA1/2* carriers without previous RRSO. Four cases of invasive cancer were detected in stage I (two ovarian cancers and two fallopian tube cancers), and three cases of invasive cancer were detected in stage III (all ovarian cancers). One patient with invasive cancer stage I had slightly elevated CA125 before surgery. On the contrary, all the patients with invasive cancer stage III had elevated CA125 and abnormal findings on transvaginal ultrasound (TVUS) before the surgical procedure (all screen detected OC). False-positive women were those who had positive screening (elevated CA125 and/or abnormal TVUS) and had no invasive OC/FTC diagnosed from surgery (screen-driven BSO) [29]. 40 *BRCA1/2* carriers were false positive (17% of BSO). No symptomatic interval cancers occurred. 

*BRCA1/2* carriers with STICs underwent RRSO only (with curative effects). Women with OC/FTC stage I were diagnosed from RRSO/screen-driven BSO. Then, they underwent additional surgery (surgical staging for OC/FTC) and were treated with adjuvant chemotherapy. The surgical staging consisted of the following: hysterectomy, peritonectomy, omentectomy, appendectomy, lymphadenectomy (pelvic, paraaortic, paracaval), and lavage with cytology of the abdominal cavity. Women with OC stage III and positive screening (elevated CA125 and abnormal TVUS) were treated with primary debulking surgery and adjuvant chemotherapy. With the median follow-up of 4.1 years, two patients died due to the progression of primary cancer and one patient was treated for a relapse of the disease.

## 4. Discussion

This is the first study in the Czech population that brings information about 496 healthy *BRCA1/2* carriers during the twenty years surveillance period. The primary endpoint of our study was to analyze the uptake of RRM and RRSO for female healthy *BRCA1/2* carriers. Our results are in wider concordance with published data. In 2019, Metcalfe et al. published a study focused on international trends in the uptake of cancer risk reduction strategies in women with P/LPVs. In a cohort of 3413 healthy *BRCA* carriers, RRM uptake was 27.8%. In comparison to the results of the Metcalfe study, the RRM uptake in our study (32.9%) was slightly higher than the uptake in neighboring Austria (28.2%), in Holland (32.7%) and much higher than the rate in Italy (10%) and Poland (4.5%). On the other hand, the rate of RRM uptake observed in our study was lower than the uptake in Norway (42.8%), the United States (49.9%) and Canada (38.0%) [22]. In 2021, Evans et al. published a prospective study of 479 women with P/LPVs in twenty years period in the United Kingdom (UK) with an uptake of RRM of 47.7% [30].

The uptake of RRSO was 48.9% in our study. In 2020, Stjepanovic et al. published a study with 853 healthy *BRCA1/2* carriers from Spain and the USA. The authors reported 41% uptake of RRSO in healthy *BRCA1/2* carriers. [18] In 2021, Ložar et al. published data on 346 *BRCA1/2* carriers followed from 1999 to 2019 in Slovenia. Only 5.8% of women underwent both preventive surgical procedures (RRM + RRSO). This is considerably lower than the uptake of both preventive surgical procedures in our study (20.8%) [31].

Another goal of our study was to analyze the uptake of prophylactic surgeries in two time periods. A significant increase in RRM count was observed in the period 2013–2020 in comparison to the period 2005–2012. In total, 12% vs. 31% of *BRCA1/2* carriers underwent RRM before and after 2013. We observed similar trends in RRSO uptake, which increased from 29% to 42%. The change in the trend in prophylactic surgery uptakes is caused by several reasons, such as the Angelina Jolie effect and progress in healthcare systems worldwide. In 2013, the famous actress Angelina Jolie disclosed to be *BRCA1* carrier and intention to undergo prophylactic surgeries. This led to widespread public discussion on this critical topic. It resulted in a substantial increase in healthy women’s genetic testing and the uptake of RRM and RRSO in healthy *BRCA1/2* carriers [32,33,34,35]. 

In a comparison of patient age, Czech *BRCA1/2* carriers underwent RRM at a younger age than the women described in the study of Metcalfe et al. (39.3 vs. 41.8 years) [22]. There is no official recommendation regarding the age of RRM in *BRCA1/2* carriers. Nevertheless, RRM performed at a younger age reduces the cumulative lifetime risk of BC more significantly than at higher age [36]. The final decision to undergo or not to undergo surgical intervention should be a personal choice of a well-informed woman. We noticed the trend of an increased mean age of women at the time of RRM underwent before and after 2013 (39.2 vs. 40.2 years); the difference in results was not statistically significant.

The mean age of Czech women at the time of RRSO was lower (43.2 years) in comparison to *BRCA1/2* carriers (45.6 years) in the study of Metcalfe et al. [22]. RRSO is recommended once the desire for pregnancy is completed in women with *BRCA1* P/LPVs from the ages of 35 to 40 years, and in women with *BRCA2* P/LPVs from the ages of 40 to 45 years onwards [9,11]. Our results are consistent with these recommendations in *BRCA2* carriers who underwent RRSO at the mean age of 43.2 years. Vice versa, the mean age of our *BRCA1* carriers is higher (43.8 years) than recommended. We also described an increasing age at the time of RRSO (41.3 vs. 44.9 y) when we compared 2005–2012 with 2013–2020. The factors associated with Increased age of RRSO uptake are unclear. Our suggestion to explain this trend is that Czech women give birth to a child at higher age over time (first child’s birth, 27.5 years in 2001 vs. 30.2 years in 2020) [37]. 

Out of 220 premenopausal *BRCA1/2* carriers with RRSO, 68 women (31%) were treated with short-term hormone replacement therapy (HRT). Women who underwent both surgeries (RRSO and RRM) were more willing to take short-term HRT (41 out of 103 women, 39.8%). Manchanda et al. recommended in their scientific paper to offer HRT until the age of natural menopause (51 years) if not contraindicated (i.e., personal history of venous thromboembolism or breast cancer). Short-term HRT for *BRCA 1/2* carriers after premenopausal RRSO does not increase breast cancer risk. However, women should be counseled on HRT’s benefits and risks to make their own well-informed decisions [38].

Furthermore, we assessed the effectiveness of prophylactic surgeries (RRM, RSSO) on cancer incidence. We showed that RRM is a highly effective strategy for reducing the risk of BC. With a median follow-up of 6.0 years, only one woman was diagnosed with BC after RRM (0.6%). This was probably due to residual breast tissue after the preventive surgery. In contrast, 54 women out of 340 (15.9%) *BRCA1/2* carriers without RRM were diagnosed with BC. In 2010, Domchek et al. published a prospective international study regarding 2482 female healthy *BRCA1/2* carriers with a median follow-up of 3.5 years. No BC was diagnosed in *BRCA1/2* carriers with RRM [39]. On the contrary, 98 (7.1%) women were diagnosed with BC in the group of 1372 *BRCA1/2* carriers without RRM. Studies with longer follow-ups are needed to assess the impact of RRM on BC incidence in the long term perspective. Similarly, we showed a one hundred percent effect of RRSO on the incidence of OC. Premenopausal RRSO causes an immediate onset of menopause, which can bring the physical and emotional symptoms of natural menopause. There is also an elevated risk of osteoporosis and cardiovascular disease in the long term [40]. These side effects are additionally monitored and treated in cooperation with other specialists from our hospital. 

The surveillance program and prophylactic surgeries are fully covered by public health insurance in the Czech Republic. The surveillance program consists of clinical and radiological screening methods (MMG, MRI) combined in six months intervals. All women decided for RRM underwent breast radiologic examination maximally three months before prophylactic surgery. The effectiveness of the screening program is seen in the low number of surgically detected occult BC. Among 156 RRM, we described only three (1.9%) occult BC during the prophylactic procedure. Ložar et al. detected five cases of occult breast cancer out of 87 RRM (5.7%) [31]. More sophisticated methods of RRM have been used in healthy *BRCA 1/2* carriers in recent years. There is growing evidence that nipple-sparing mastectomy is oncologically safe, optimizes cosmetic results, and leads to higher levels of psychosocial and sexual well-being in women [41,42,43]. 

*BRCA1/2* patients with the diagnosis of BC during the surveillance are surgically treated based on the clinical stage of BC and also according to patients’ preferences. The impact of contralateral RRM on prognosis in *BRCA1/2* breast cancer patients depends on the stage of detected BC (secondary contralateral BC vs. generalization of primary BC) [44]. *BRCA1/2* breast cancer patients at a younger age are more willing to undergo bilateral mastectomy because they have a longer life expectancy and prefer the maximum reduction of BC risk. 

The treatment strategy of BC patients in our study was the following. All 5 patients with DCIS were treated with breast surgery only. Out of 50 patients with invasive BC, 45 women (90%) underwent surgery with chemotherapy, and 5 women (10%) underwent surgery with endocrine therapy. 32 women (52.8%) with screen-detected BC decided for bilateral mastectomy. All 4 patients with STIC underwent RRSO only, and all 7 patients (100%) with invasive OC/FTC were treated with surgery and chemotherapy.

Our study is unique because of its long period of surveillance from 2000 to 2020. The median follow-up of 6.0 years is longer compared to previously published literature. However, studies with more extensive follow-ups are still needed to evaluate the effectiveness of prophylactic surgeries in the long term perspective. A large statistical set of 496 healthy female *BRCA1/2* carriers belongs to the most extensive study in the region of Central Europe. The study’s strengths are the good consistency of collected data and the high compliance of *BRCA1/2* carriers in the surveillance program. Only 2.6% of healthy *BRCA1/2* carriers were excluded from the study due to incomplete records. A limitation of the study is the retrospective character of the data. In the future, we plan to collect prospective data from several institutions in the Czech Republic to obtain other important information for improving the care of *BRCA1/2* healthy carriers.

## 5. Conclusions

The study brings information regarding the healthy *BRCA1/2* population in Central Europe. The study demonstrates the effectiveness of RRM on the incidence of BC and RRSO on the incidence of OC in healthy *BRCA1/2* carriers. Presented data are relevant not only for the decision-making process of experts but also for illustration and counseling of cancer risk to healthy female *BRCA1/2* carriers. The results may be used in public healthcare efforts on the national and international levels.

## Figures and Tables

**Figure 1 cancers-15-01072-f001:**
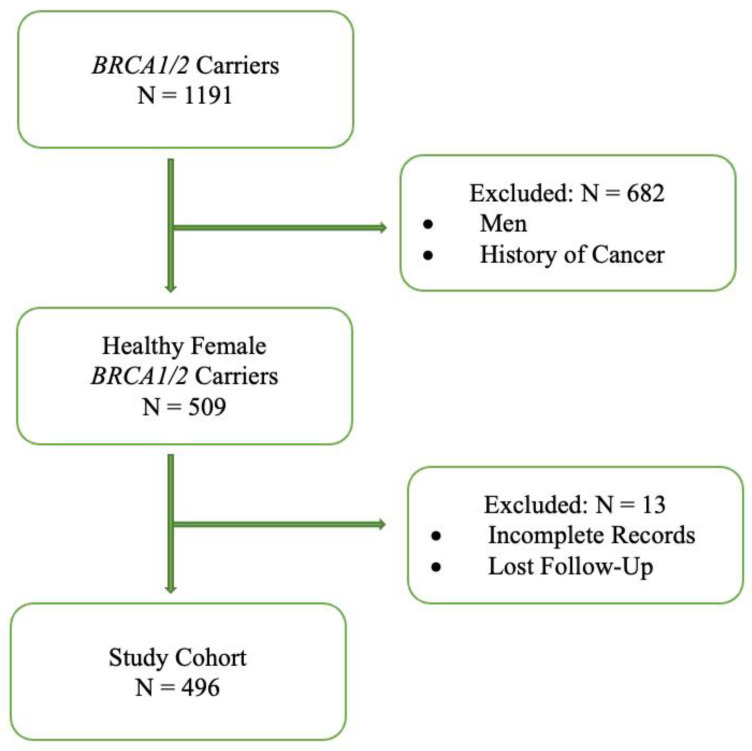
Flow-Chart: Selection of Study Population.

**Figure 2 cancers-15-01072-f002:**
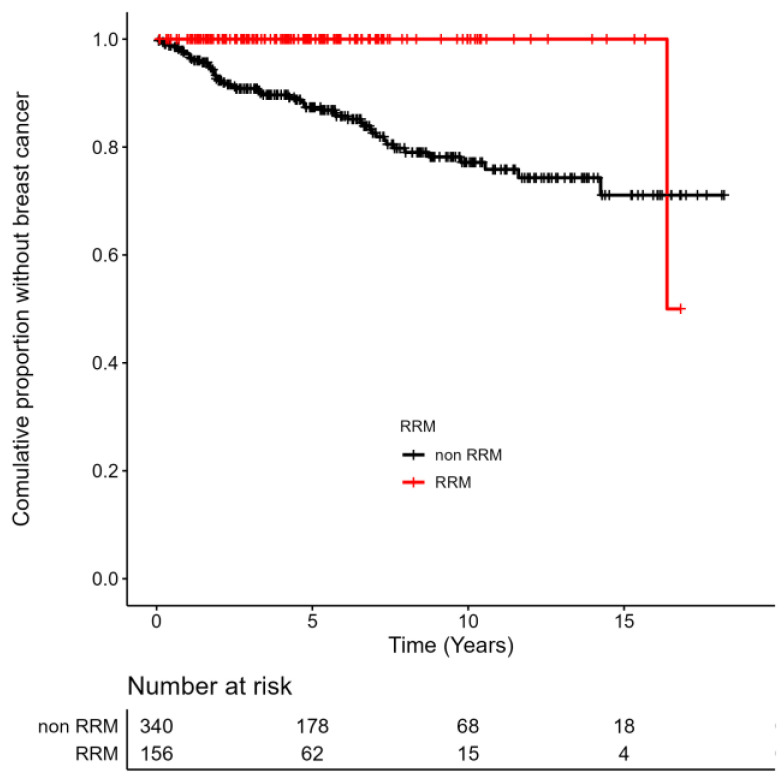
Incidence of Breast Cancer (BC) in *BRCA1/2* Carriers With and Without RRM (Kaplan-Meier Curves): In the cohort of *BRCA1/2* carriers with RRM, the time was calculated from the RRM till the occurrence of BC or the end of the study in 2020. In the cohort of *BRCA1/2* carriers without RRM, the time was calculated from the initiation of surveillance of the woman till the occurrence of BC or the end of the study in 2020. The term “complete” represents *BRCA1/2* carriers with diagnosed BC. The term "censored" represents the *BRCA1/2* carriers without BC. RRM—risk-reducing mastectomy. A significant result was achieved, HR 20.18, Log-Rank test, *p* < 0.001.

**Figure 3 cancers-15-01072-f003:**
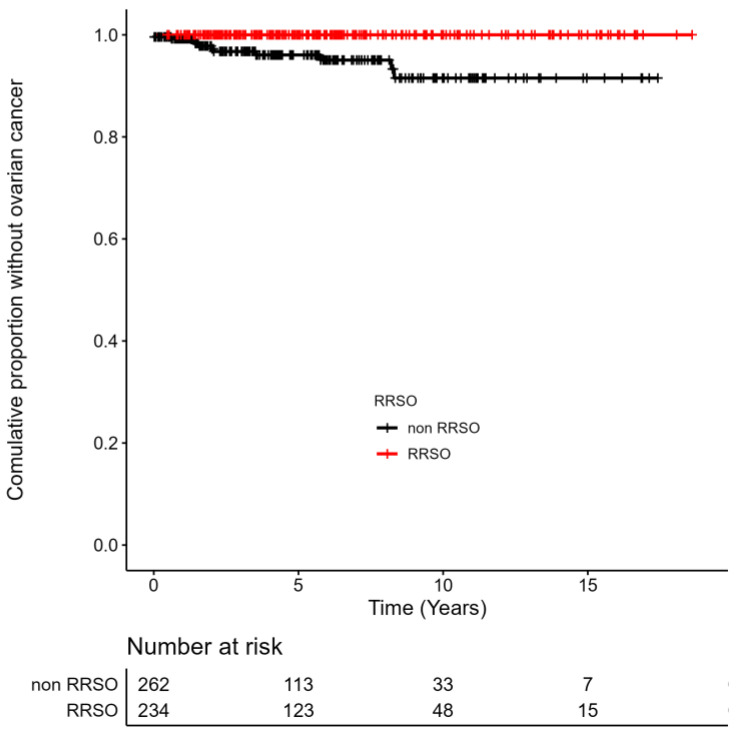
Incidence of Ovarian Cancer (OC) in BRCA1/2 Carriers With and Without RRSO (Kaplan-Meier Curves): In the cohort of *BRCA1/2* carriers with RRSO, the time was calculated from the RRSO till the occurrence of OC or the end of the study in 2020. In the cohort of *BRCA1/2* carriers without RRSO, the time was calculated from the initiation of surveillance of the woman till the occurrence of OC or the end of the study in 2020. RRSO—risk-reducing oophorectomy. A significant result was achieved; HR was not defined due to 0% occurrence of OC in the RRSO group, Log-Rank test, *p* < 0.001.

**Table 1 cancers-15-01072-t001:** *BRCA1/2* Carriers Surveillance Data and Prophylactic Procedures (RRM, RRSO, Combination of RRM and RRSO).

	Total	RRM (Only)	RRSO (Only)	RRM + RRSO (Combined)	Surveillance (Only)
*BRCA1/2* Carriers, N (%) *BRCA1*, N (%) *BRCA2*, N (%)	496 (100) 348 (100) 148 (100)	53 (10.7) 40 (11.5) 13 (8.8)	131 (26.4) 91 (26.1) 40 (27.0)	103 (20.8) 71 (20.4) 32 (21.6)	209 (42.1) 146 (41.9) 63 (42.6)
Mean Age, y (range) *BRCA1*, y, (range) *BRCA2*, y, (range) Median Age, y	35.4 (18–65) 35.2 (18–65) 35.9 (18–60) 35.0	32.9 (22–60) 33.1 (22–56) 32.4 (26–60) 32.0	43.2 (28–64) 43.8 (28–64) 43.2 (34–57) 42.0	- - - -	- - - -
Mean Follow-up (range) Median Follow-Up, y	6.8 (0.5–20) 6.0	6.6 (0.5–20)	7.9 (0.5–16)	-	5.4 (0.5–17)

RRM—risk-reducing mastectomy. RRSO—risk-reducing salpingo-oophorectomy. y—years. %—described in a row.

**Table 2 cancers-15-01072-t002:** Prophylactic Surgeries in *BRCA1/2* Carriers in Different Time Periods.

	Total	RRM	RRSO
**2005–2012** - Recruited Women, N (%) - Prophylactic Surgery (PS), N (%) - TTPS, y - Mean Age at PS, y	192 (100) - - -	- 23 (12.0) 2.3 39.2	- 56 (29.2) 1.5 41.3
**2013–2020** - Recruited Women, N (%) - Prophylactic Surgery, N (%) - TTPS, y - Mean Age at PS, y	304 (100) - - -	- 96 (31.6) 1.8 40.2	- 129 (42.4) 1.0 44.9

RRM—risk-reducing mastectomy. RRSO—risk-reducing salpingo-oophorectomy. TTPS—time to prophylactic surgery. y—years.

**Table 3 cancers-15-01072-t003:** Characteristics and Treatment of Breast Cancer (BC) in BRCA1/2 Carriers.

	Total *BRCA1/2* N = 496	*BRCA1* N = 348	*BRCA2* N = 148
Breast Cancer, N (%)	55 (11.1)	46 (13.2)	9 (6.1)
Histology, N (%) - DCIS - IDC - ILC	5 (9.1) 49 (89.1) 1 (1.8)	4 (8.7) 41 (89.1) 1 (2.2)	1 (11.1) 8 (88.9) 0
A subtype, N (%) - TNBC - ER-positive/HER2-negative - ER-positive/HER2-positive - ER-negative/HER2-positive	29 (52.7) 23 (41.8) 2 (3.7) 1 (1.8)	29 (63.1) 15 (32.7) 1 (2.1) 1 (2.1)	0 8 (88.9) 1 (11,1) 0
Grade, N (%) - 1 - 2 - 3	3 (5.5) 15 (27.3) 37 (67.2)	3 (6.5) 10 (21.7) 33 (71.8)	0 5 (55.6) 4 (44.4)
Invasive Breast Cancer-Stage, N (%) - I - II - III–IV	27 (54.0) 23 (46.0) 0	22 (52.4) 20 (47.6) 0	5 (62.5) 3 (37.5) 0
Detection, N (%) - Screen Detected BC - Interval BC	42 (76.3) 13 (23.7)	34 (73.9) 12 (26.1)	8 (88.9) 1 (11.1)
The Sequence of Treatment, N (%) - Primary Surgery Only - Primary Surgery→ET only - Primary Surgery→ACT - NACT→Surgery	5 (9.1) 5 (9.1) 25 (45.4) 20 (36.4)	4 (8.7) 3 (6.5) 21 (45.7) 18 (39.1)	1 (11.1) 2 (22.2) 4 (44.5) 2 (22.2)
Type of Breast Surgery, N (%) - Breast-Conserving Surgery - Unilateral Mastectomy - Bilateral Mastectomy	18 (32.7) 5 (9.1) 32 (58.2)	12 (26.1) 5 (10.9) 29 (63.0)	6 (66.7) 0 3 (33.3)

DCIS—ductal carcinoma in situ. IDC—invasive ductal carcinoma. ILC—invasive lobular carcinoma. TNBC—triple-negative breast cancer. ER—estrogen receptor-positive BC, HER2—epidermal growth factor receptor type 2 -positive BC, NACT—neoadjuvant chemotherapy. ACT—adjuvant chemotherapy. ET—endocrine therapy.

**Table 4 cancers-15-01072-t004:** Characteristics, Diagnosis, and Treatment of Ovarian Cancer (OC) in *BRCA1/2* Carriers.

Patient	*BRCA1/2*	Diagnosis	Ca125	TVUS	Surgery	CT	FU (Months)	Alive/ Deceased
1	*BRCA1*	STIC	Normal	Normal	RRSO	No	30	Alive
2	*BRCA1*	STIC	Normal	Normal	RRSO	No	34	Alive
3	*BRCA1*	STIC	Normal	Normal	RRSO	No	79	Alive
4	*BRCA1*	STIC	Normal	Normal	RRSO	No	97	Alive
5	*BRCA1*	FTC, st. I	Normal	Normal	SS	Yes	32	Alive
6	*BRCA2*	FTC, st. I	Normal	Normal	SS	Yes	49	Alive
7	*BRCA1*	OC, st. I	Elevated	Normal	SS	Yes	17	Alive
8	*BRCA1*	OC, st. I	Normal	Normal	SS	Yes	81	Alive, relapse
9	*BRCA1*	OC, st. III	Elevated	Abnormal	PDS	Yes	20	Alive
10	*BRCA2*	OC, st. III	Elevated	Abnormal	PDS	Yes	55	Deceased
11	*BRCA1*	OC, st. III	Elevated	Abnormal	PDS	Yes	73	Deceased

STIC—Serous tubal intraepithelial cancer. FTC—Fallopian tube cancer. OC—Ovarian cancer. TVUS—Transvaginal ultrasound. SS—Surgical staging. PDS—Primary debulking surgery. CT—Chemotherapy. FU—Follow-up, from diagnosis till the end of the study or patient’s death.

## Data Availability

Not applicable.

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
