# Peer review of "Uptake and Effectiveness of Risk-Reducing Surgeries in Unaffected Female BRCA1 and BRCA2 Carriers: A Single Institution Experience in the Czech Republic"

_cancers, 2023, doi:10.3390/cancers15041072_

Round 1

Reviewer 1 Report

This is a well-written paper on an important topic. Thanks for the opportunity to review it. I have quite a few comments, but all are minor.

Introduction:

-          What does “The amount of hereditary-related BRCA1/2 gene alterations is 8-13%” mean? Do you mean 8-13% of people with BC have BRCA 1/2 gene alterations?

-          “All healthy BRCA1/2 carriers in the Czech Republic take part in the specific preventive program” It would be useful to define healthy (I assume you mean without a prior cancer diagnosis). People can be unhealthy and not have cancer, so I’d suggest changing the wording. Can you also describe the Czech Republic program briefly? Is it really “all” carriers? Or just all carriers at the one hospital in Prague that is reported on?

Methods:

-          “The final decision to undergo or not to undergo surgical intervention should be a personal choice of a well-informed woman.” This sentence is better suited for the discussion, as it’s not a method.

-          Consider using more inclusive language. Trans men can (and do) have BRCA 1/2 pathogenic mutations. You note you excluded men in the results, which I assume means individuals assigned male at birth? That should be in the methods instead of the results.  

-          Why was 2013 decided as the cutpoint for the year groupings? Given the change in testing, it seems like 2015 might be a more intuitive point.

Results:

-          “mutations” shouldn’t be used alone when you mean “pathogenic mutations.” Everyone has mutations, but they don’t all have clinical significance.

-          Table 1. Would be helpful to note that percentages are row percents.

-          Table 2. The formatting seems to be off, because the row with recruited women doesn’t have any data in it.

-          Section 3.3 “The residual breast tissue was found in this patient after the RRM (16 years ago), which was probably the reason for the development of BC.” This belongs in the discussion, as it is an opinion.

Discussion:

-          Metcalfe didn’t publish as a sole author, so it should say Metcalfe et al. and then “they” (not she). This should be corrected every time Metcalfe is referenced. Note a type that says Metcalfe s’

-          I’d like to see more discussion about why you think your population is different in the percent of carriers having RRM and RRSO compared to the groups reported by Metcalfe et al.

-          Non-BC I think should be no BC?

-          “Up to 80% of BC patients and 64% of OC 366 patients were treated with CT with considerable side effects and a decrease in the woman's quality of life.” I don’t think you measured side effects and QOL. You likely mean “which can have considerable side effects and cause a decrease in quality of life.”

-          “Brave” does not seem to be an appropriate term to use for younger women who get a bilateral mastectomy. Please hypothesize another reason for this. I suggest that younger women have additional years to develop BC, so they might be more willing to have more dramatic surgery. Or they are less likely to have other health issues impacting surgery? Or they might have more access to genetic information and resources?

-          Limitations – could surgeries have happened that you were unable to capture because they happened outside of the Czech Republic?

-          “The uptake of RRM and RRSO in the Czech Republic belongs to the better European average.” There is a grammatical error here, so I’m not sure what you’re trying to say. Do you mean it’s higher than most other European countries?

Minor comments:

-          In the abstract, line 31, “in” is missing in “BC developed 15.9%”

-          Section 2.1 line 109 I think has a typo – RRME instead of RRM

-          Section 3.1 line 155, RRM underwent 111… should be reworded. I think you mean RRM was undertaken…

-          Section 3.3 line 196 should say “in total” rather than “totally”

-          Page 7 line 238-239 isn’t grammatically correct.

-          Table 3 has an errant % in the grade 3 section and the formatting is off for the data in sequence of treatment

Author Response

Dear reviewer,

Thank you very much for your excellent and meaningful review. I am sending you the information about how we adjusted the text according to your suggestions in the attachment. 

Very respectfully
Martina Zimovjanova

Reviewer 2 Report

Thank you for asking me to review this manuscript. Zimovjanova et al. report a single-centre, retrospective study outlining the prevalence of breast and ovarian cancer in unaffected germline BRCA1/2 heterozygotes following RRM or RRSO.

Major comments

1. Strengths: large data set (496), good follow-up (median 6 years), thorough analysis of data.

2. Weakness: retrospective study (cannot be helped), no new findings other than data from Central Europe. The discussion is written relatively poorly – the sentence structure and use of English could be improved. The latter point is the main criticism of this manuscript.

Minor comments

1.     If BRCA1 or BRCA2 are written in italics, there is no need to write ‘BRCA1 gene’ instead just ‘BRCA1’ will suffice.

2.     Staging of BC does not include stage 0 (DCIS). Please delete.

3.     Table 3 – were there any ER-positive/HER2-positive cases?

4.     There is no such thing as ‘FIGO stage 0’ OC as shown in Table 4. Please delete

5.     STIC is not a form of OC as alluded to Table 4. Please delete.

6.     The ‘primary surgery only’ n=4 were the STIC cases. STIC is not a form of ovarian cancer and therefore is not treated with chemotherapy, therefore remove row

7.     Table 4 – BRCA1 and BRCA2 should be written in Italics.

8.     Section 3.5 is irrelevant and can be deleted.

9.     Discussion: ‘She reported’ should be ‘Metcalfe reported’ or ‘Their group reported’

10.  Discussion: BRCA1 and BRCA2 should be written in Italics when describing the genes.

11.  Discussion: unusual way to start a sentence with the word ‘Totally’

12.  Discussion: ‘Angelina Jolie’s revelation’… I would avoid the word ‘revelation’ and also would state what her revelation was i.e. of being a germline BRCA1/2 heterozygote.

13.  Discussion: the following sentence makes no sense and is meaningless. ‘Different healthcare systems in other countries, the impact of the region's socioeconomic situation, cultural aspects, and the number of family members with BC/OC may have a decisive impact on the uptake of prophylactic surgeries. We did not analyze these aspects in our dataset.’ This can be deleted.

14.  Discussion: ‘Vice versa, the mean age of our BRCA1 carriers is higher (43.8 years) as recommended’… should be ‘higher THAN recommended'

15.  Discussion: ‘Up to 80% of BC patients and 64% of OC patients were treated with CT with considerable side effects and a decrease in the woman's quality of life.’ Authors did not include side effect or QoL data and therefore should delete sentence.

16.  Discussion: Did RRSO surgical practice change over 20 years?

17.  Conclusion: ‘the study brings unique’… I would not say this is ‘unique’ information

18.  Conclusion: this sentence does not make sense: “The uptake of RRM and RRSO in the Czech Republic belongs to the better European average’.

19.  Conclusion: this sentence does not make sense: ‘In sum with the significant increase in prophylactic procedures in recent years, it is a good signal for the next period’.

Author Response

(The authors gave the same response as above.)
